# Evaluating Standardized Job Competencies for Managing Students at Risk for Anaphylaxis in Elementary School Nurses

**DOI:** 10.3390/healthcare11142102

**Published:** 2023-07-24

**Authors:** Yeon-Ha Kim, Aee Lee Kim

**Affiliations:** 1Department of Nursing, Korea National University of Transportation, Jeungpyeong-gun 27909, Republic of Korea; tiffany7@ut.ac.kr; 2College of Nursing, Sungshin Women’s University, Seoul 01133, Republic of Korea

**Keywords:** anaphylaxis, school nursing, primary school, emergency nursing, work performance

## Abstract

This study aimed to evaluate standardized job competencies of elementary school nurses in managing the health of students at risk for anaphylaxis (SRAs). A total of 166 elementary school nurses from across Korea participated in this study. The tool utilized was a list of standardized job tasks of elementary school nurses in managing SRAs’ health. Seven factors were obtained from the factor analysis, with a cumulative variance explained of 68.6%. The importance–performance analysis method was employed to suggest priority areas for training. The factors placed in quadrant II included: (1) “offering psychological support”, among elementary school nurses who have been provided with clinical information on anaphylaxis, (2) “providing emergency care”, among those who have not been provided with clinical information on anaphylaxis, (3) the factors “utilizing individualized healthcare plans”, “conducting health education and staff training”, and “evaluating the student”, among those who agreed that schools need legal protection to purchase epinephrine auto-injectors, and (4) the factor “preparing a school emergency system”, among those with less than five years of experience as a school nurse. When planning continuing education strategies to improve elementary school nurses’ management of SRAs, factors such as career experience, access to clinical information, and beliefs regarding legal protection for schools to purchase epinephrine auto-injectors should be considered.

## 1. Introduction

Anaphylaxis is a severe, potentially life-threatening allergic reaction that has become a significant health issue in schools. It is estimated that approximately 3 million children under the age of 18 years have reported experiencing a food allergy reaction in the past 12 months [1]. Among children receiving medical treatment, 30.5% experienced anaphylaxis, and of those, 52.6% were of elementary school age [2]. The most common causes of anaphylaxis include allergies to food, insects, medication, latex, and other factors, which can result in severe reactions such as airway obstruction, swelling, dyspnea, cyanosis, and skin symptoms [3]. Elementary students with severe allergies in the school system pose a serious concern for schools regarding their ability to appropriately respond to anaphylactic events and ensure the safety of these students [4]. Considering the challenges of engaging students in taking an active role in managing their health issues, school nurses are considered the most qualified professionals to coordinate and implement care for these children at school [4,5].

Elementary school nurses face challenges in identifying and managing anaphylaxis in the school setting. In the United States, a school nurse-to-student ratio of 1:750 has been widely recommended [5], yet workload management remains inadequate to address the increasingly complex health needs of students [6]. In contrast, in South Korea, any elementary school with at least 18 classes must employ one school nurse, and as of December 2021, those with at least 36 classes must employ two school nurses [7], resulting in a ratio of 1:1000 [8]. Consequently, elementary school nurses in South Korea are responsible for larger numbers of students and face particularly intense job demands [9]. Since they play a crucial role in helping children avoid allergens and providing appropriate care when allergic reactions occur, there is a need to improve their competency. Training programs for school nurses have been conducted in other countries to improve on responding emergencies of students with life-threatening allergies [10,11,12], and manuals have been distributed to serve as guidelines [13,14,15]. In South Korea, although school nurses are now legally allowed to inject epinephrine with auto-injectors in a school setting [8], a specific training curriculum of anaphylaxis for school nurses is lacking. Nurses encounter anaphylaxis in the context of general health education, but standardized manuals have yet to be developed and distributed. Therefore, many of them possess varying levels of knowledge and experience, which can lead to gaps and limitations in managing anaphylaxis in the school setting [10,16].

To overcome these limitations, standardized job competencies are necessary for school nurses to perform health management of students at risk for anaphylaxis (SRAs). Job competencies consist of individual steps of job tasks that form specific, meaningful units of work required for a given job [17]. It is essential to assess how elementary school nurses perform standardized job competencies and perceive their importance and difficulty so that they can be incorporated into continuing education (CE) programs. This effort could help reduce the gap in elementary school nurses’ management of SRAs and enhance the quality of health services.

To determine the priority areas for elementary school nurses in managing SRAs, importance–performance analysis (IPA) can be a valuable method [13]. This approach aids in identifying problematic areas for strategically developing training curricula and improving the quality of health services [18].

In previous studies, most anaphylaxis research in school nursing focused on the evaluation of food allergies [10], knowledge and attitudes towards anaphylaxis [16,19], school policies for food allergies and anaphylaxis [10,20,21,22], and health education and staff training [11,23]. However, to the best of our knowledge, no prior studies have assessed the standardized job competency of health management of SRAs for elementary school nurses and identified areas requiring training. Therefore, the aim of this study was to evaluate the health management of SRAs as practiced by elementary school nurses based on standardized job competencies and suggest priority areas for training. This study may be valuable, as its results could be used as foundational data for developing a systematic, evidence-based training curriculum on the health management of SRAs for elementary school nurses.

## 2. Materials and Methods

### 2.1. Participants and Procedures

In this study, a cross-sectional survey was conducted that focused on the standardized job competencies of elementary school nurses in managing the health of SRAs.

In total, 166 elementary school nurses from across the nation participated in this study. The inclusion criteria for the elementary school nurses were as follows: (1) managing the health of students at risk for anaphylaxis, (2) having worked as an elementary school nurse for over a year, (3) working in a full-time position, and (4) voluntary participation in this study. The exclusion criteria for elementary school nurses included: (1) managing general allergic reactions or asthma, rather than hypersensitivity reactions, (2) having worked as an elementary school nurse for less than a year, (3) working in a part-time capacity, and (4) holding a short-term teaching position.

The sample size was determined using the G*Power 3.1 program (Heinrich Heine University, Düsseldorf, Germany), employing correlation analysis with an effect size of 0.2, a statistical power of 80%, and a significance level of 0.05, while accounting for an anticipated dropout rate of 20%. The minimum sample size was estimated to be 160; thus, the number of participants was deemed to be appropriate.

### 2.2. Measurements

The tool used in this study was developed by Kim (2020), with the aim of providing standardized job tasks for elementary school nurses in managing the health of SRAs [24]. This tool was created using the job analysis method, involving seven focus group members (4 school nurses, 1 emergency room nurse, 1 university professor in nursing, and 1 nurse with expertise in job analysis) who were experts in the field of school health. These members convened 10 times for 180 minutes each to develop job tasks associated with the elementary school nurses’ management of the health of SRAs based on consensus [17]. Subsequently, an examination process was carried out based on manuals obtained from the Ministries of Education in South Korea, the United States, and Canada [13,14,15]. This process aimed to verify the accuracy of the identified job tasks and to ensure they were not based on opinions or biases. Through this process, 96 tasks were determined to be essential for the role of an elementary school nurse in managing the health of SRAs.

In this study, a self-report questionnaire was constructed to assess the performance, importance, and difficulty level of various job tasks. Participants rated each item (task) on a 5-point Likert scale, ranging from 5 (very highly performed, very highly important, very highly difficult) to 1 (minimally performed, minimally important, minimally difficult). The questionnaire also included demographic variables of the school nurse, such as age, working experience as a hospital nurse, working experience as a school nurse, number of classes, previous anaphylaxis events in the community, access to clinical information, and agreement on whether schools should be legally allowed to purchase epinephrine auto-injectors.

Although the selected items had already been verified through the consensus of the focus group members [17], content and construct validity were assessed in this study using responses from 166 school nurses who were currently managing the health of students at risk for anaphylaxis. Fourteen items that did not meet the standard content validity index (0.80) were removed. To confirm the underlying structure of the rest of the items, factor analysis was conducted, resulting in the extraction of seven factors, with a cumulative variance explained of 68.6%. No further items were eliminated. Consequently, 82 items of standardized job competency for elementary school nurses in managing the health of students at risk for anaphylaxis were developed. The seven factors obtained from the factor analysis were renamed as follows, based on manuals [13,14,15] and previous research [24]: (F1) assessing priority issues, (F2) utilizing individualized healthcare plans, (F3) preparing a school emergency system, (F4) providing emergency care, (F5) offering psychological support, (F6) conducting health education and staff training, and (F7) evaluating the student. The internal consistency of the performance, importance, and difficulty of this tool had Cronbach’s alpha values of 0.982, 0.976, and 0.989, respectively.

### 2.3. Data Analysis

All statistical analyses were performed using SPSS version 23.0 (IBM Corp., Armonk, NY, USA). Differences in performance levels based on participant characteristics were identified using the t-test and analysis of variance. Post hoc tests were conducted using Duncan’s method. Standardized job competencies for suggesting training needs were determined through determinant coefficients obtained by multiplying the mean values of importance and difficulty levels, which ranged from 1 to 25. Correlations between the performance, importance, and difficulty of factors were analyzed using Pearson’s correlation coefficients. To identify the predictive variables for performance levels in managing the health of SRAs, stepwise linear regression was performed. Independent variables (working career as a school nurse, providing clinical information, allowing schools to purchase epinephrine auto-injectors under the law) that demonstrated significant differences in performance levels were converted to dummy variables and entered. Factors (importance of F1~F7) that exhibited correlations with performance levels were also included.

Using the IPA method, we identified problematic areas among seven factors that should be prioritized for training. A two-dimensional IPA model was created, with a mean importance level of 4.46 and a mean performance level of 4.01. The model was divided into four quadrants, with performance on the x-axis and importance on the y-axis. Quadrant I contained attributes that were perceived as highly important with a high level of performance. School nurses should be encouraged to maintain their excellent work in these areas. Quadrant II included factors that were considered very important, but with a relatively low performance level. Improvement efforts should be focused on enhancing performance in these areas. Quadrant III consisted of attributes that were perceived as having low importance and low performance. These factors should be considered a low priority for training. Lastly, quadrant IV contained attributes of low importance but relatively high performance. In these situations, some attributes may be overemphasized, indicating that current practices may be unnecessary [18].

### 2.4. Data Collection

In South Korea, there are 17 offices of education distributed across the country and 4562 elementary school nurses, as of 2020. Each region has a branch of the Korean Health Teachers Association. The authors contacted each branch and informed them of the study’s purpose. Each branch then recruited participants who had experience managing students at severe risk of anaphylaxis, using their existing SNS networks. In total, 270 school nurses volunteered for this study. The authors called each school nurse to confirm whether they would participate in the study and mailed questionnaires to 240 elementary school nurses who met the inclusion criteria. The authors also called each school nurse to remind them to respond to the questionnaire. A total of 210 elementary school nurses returned the questionnaire, and after excluding missing data, 166 participants were included in this study. The dropout rate was 20%. Data collection took place from January to May 2021.

### 2.5. Ethical Considerations

Each school nurse was informed of the study’s purpose and their right to withdraw without penalty through a phone call. Before providing written consent, the nurses were assured of confidentiality and anonymity. The Institutional Review Board (IRB) of the researchers’ affiliated university approved this study (IRB No. KNUT IRB. 2020-07). The authors contacted each participant individually by phone to determine their eligibility based on the inclusion and exclusion criteria.

## 3. Results

### 3.1. Differences in Performance Level Based on Participants’ Characteristics

Table 1 shows the characteristics of the study participants. The majority (79.6%) of the subjects were over 40 years old, with a mean age of 45.77 ± 8.85 years. Most participants (71.7%) graduated university. Most participants (60.2%) had more than three years of experience working as a hospital nurse. Among the subjects, 17.5% had less than five years of experience as a school nurse and demonstrated a significantly lower performance level compared to those with more than five years of experience (t = 2.85, *p* = 0.039). The average number of classes they were responsible for was 25.87 ± 13.61. The average length of their school nursing career was 16.41 ± 9.64 years. A total of 42.8% of the subjects had previously experience with events of anaphylaxis in the community. The majority (94.6%) of participants received clinical information through leaflets (9.0%), CE courses (71.7%), and the internet (13.9%). These individuals exhibited a significantly higher performance level (t = 2.28, *p* = 0.023) compared to those who did not receive such information. Subjects who agreed that schools should be legally allowed to purchase epinephrine auto-injectors (43.4%) demonstrated a significantly higher performance level (t = 2.14, *p* = 0.033) compared to those who disagreed.

### 3.2. Performance Level and Determinant Coefficients

The factor with the highest determinant coefficients was “F4: providing emergency care”, with a score of 16.15 ± 5.60 (Table 2). Other factors that showed relatively high coefficients of determination were “F2: utilizing individualized healthcare plans” (15.62 ± 4.64), “F1: assessing priority issues” (15.32 ± 4.99), “F6: conducting health education and staff training” (14.96 ± 4.46), “F3: preparing a school emergency system“ (14.89 ± 4.95), “F5: offering psychological support” (14.37 ± 5.48), and “F7: evaluating the student” (14.35 ± 5.45), listed in descending order. The factor that exhibited the lowest performance score was “F7: evaluating the student” (3.58 ± 0.95). Other factors with relatively low performance scores were, “F6: conducting health education and staff training” (3.60 ± 0.89), “F2: utilizing individualized healthcare plans” (3.66 ± 0.89), “F5: offering psychological support” (3.94 ± 0.90), “F1: assessing priority issues” (4.10 ± 0.80), “F3: preparing a school emergency system” (4.16 ± 0.78), and “F4: providing emergency care” (4.42 ± 0.77), listed in ascending order.

The item of job competency that showed the highest coefficient of determination was “F4-2-2: administer an epinephrine injection immediately”, with a score of 20.10 ± 5.82. Relatively high scores were also found for “F4-1-1: when the airway is clear, remove the triggering cause” (17.93 ± 6.12), “F4-2-6: lead in performing cardiopulmonary resuscitation, if needed” (17.68 ± 7.21), and “F4-5: teach SRAs how to self-administer their epinephrine medication” (17.62 ± 6.67). The job competency item that showed the lowest performance score was “F6-10: conduct staff training on administering an epinephrine auto-injector”, with a score of 3.17 ± 1.43. Relatively low scores were also found for “F6-5: prepare a health education teaching plan for students with anaphylaxis” (3.28 ± 1.22), “F2-9: collaborate with the principal, guardians, SRAs, class teachers, school nutritionists, and other related school staff to build alliances, and take the lead in these alliances” (3.37 ± 1.24), and “F7-7: evaluate whether staff training was carried out as planned” (3.40 ± 1.11). The item of job competency that showed the highest performance score was “F3-16: complete a course in cardiopulmonary resuscitation training”, with a score of 4.65 ± 0.82. Relatively high scores were also found for “F4-1-7: when the airway is clear, contact the guardian” (4.56 ± 0.89) and “F4-1-4: when the airway is clear, monitor the student until symptoms subside or until the guardian arrives” (4.54 ± 0.89).

### 3.3. Correlations between Performance, Importance, and Difficulty Level in Managing Health of Students at Risk for Anaphylaxis

The correlations between factors of performance, importance, and difficulty are shown in Table 3. Performance showed a significant positive correlation with all factors for importance: “F1: assessing priority issues” (r = 0.39, *p* < 0.001), “F2: utilizing individualized healthcare plans” (r = 0.48, *p* < 0.001), “F3: preparing a school emergency system” (r = 0.53, *p* < 0.001), “F4: providing emergency care” (r = 0.43, *p* < 0.001), “F5: offering psychological support” (r = 0.44, *p* < 0.001), “F6: conducting health education and staff training” (r = 0.44, *p* < 0.001), and “F7: evaluating the student” (r = 0.45, *p* < 0.001). There were no correlations between performance and any factors of difficulty. However, performance for F1 (assessing priority issues) showed significant correlations with difficulty for all factors.

### 3.4. Predictive Variables for the Performance Level in Managing the Health of Students at Risk for Anaphylaxis

The predictive variables for the performance level in managing the health of SRAs are shown in Table 4. We found that perceiving the importance of “F3: preparing a school emergency system” (β = *0*.52, *p* < 0.001) and working as a school nurse for less than 5 years (β = −0.13, *p* = 0.043) had significant effects on participants’ performance. These predictive variables explained 29.6% of the variance.

### 3.5. Areas That Should Be Prioritized for Training Elementary School Nurses in Managing the Health of Students at Risk for Anaphylaxis

The results of areas that should be prioritized for training school nurses are presented in Figure 1. For school nurses managing the health of SRAs, the factor “F5: offering psychological support” was included in quadrant II, while the factors “F2: utilizing individualized healthcare plans”, “F6: conducting health education and staff training”, and “F7: evaluating the student” were included in quadrant III.

For school nurses who had been provided with clinical information on anaphylaxis, the factor “F5: offering psychological support” was placed in quadrant II, while the factors “F6: conducting health education and staff training” and “F7: evaluating the student” were included in quadrant III. For school nurses who had not been provided clinical information on anaphylaxis, the factor “F4: providing emergency care” was placed in quadrant II, while the other factors were all placed in quadrant III. For school nurses who agreed that schools need to be allowed by law to purchase epinephrine auto-injectors, the factors “F2: utilizing individualized healthcare plans”, “F6: conducting health education and staff training”, and “F7: evaluating the student” were placed in quadrant II. For school nurses who disagreed that schools need to be allowed by law to purchase epinephrine auto-injectors, the factors “F2: utilizing individualized healthcare plans”, “F5: offering psychological support”, “F6: conducting health education and staff training”, and “F7: evaluating the student” were placed in quadrant III. For school nurses with less than 5 years of career experience as a school nurse, the factor “F3: preparing a school emergency system” was plotted in quadrant II, while the factors “F2: utilizing individualized healthcare plans”, “F5: offering psychological support”, “F6: conducting health education and staff training”, and “F7: evaluating the student” were plotted in quadrant III. For school nurses with over 5 years of career experience, no factors were plotted in quadrant II, while the factors “F2: utilizing individualized healthcare plans”, “F6: conducting health education and staff training”, and “F7: evaluating the student” were plotted in quadrant III.

## 4. Discussion

This study was a cross-sectional survey based on standardized job competencies of elementary school nurses in managing the health of SRAs. This study makes a meaningful contribution to the literature as the first study to investigate the competencies of elementary school nurses in managing the health of SRAs by using the IPA method and determinant coefficients to evaluate standardized job competencies. The primary aim of this study was to contribute to the development of a CE curriculum for elementary school nurses in SRAs’ management by providing essential data.

In this study, IPA revealed that the factors “assessing priority issues”, “preparing a school emergency system”, and “providing emergency care” were placed in quadrant I, which encourages elementary school nurses to keep up the good work. This result indicates that elementary school nurses play a crucial role in assessing and managing life-threatening emergencies. These factors also demonstrated the highest performance levels.

The performance of the “assessing priority issues” factor was correlated with importance and difficulty for all factors. The items of this factor consisted of: “identify SRAs through a health status survey”, ”maintain an updated list of SRAs”, “secure an action plan from the guardian, completed by a medical doctor, for health needs assessment”, “obtain written consent from guardians for sharing personal information and providing first aid”, “evaluate the self-care proficiency of SRAs”, “ascertain the present state of EMSS and alliances”, “assess the availability of anaphylaxis management resources such as medicine, equipment, and budget”, and “develop an emergency care plan”. This result suggests that elementary school nurses who perform comprehensive assessments of SRAs are more likely to perceive all areas as important. It appears that conducting a general assessment is not an easy task; therefore, higher performance requires greater caution.

Using stepwise linear regression analysis, the perceived importance of the factor “preparing a school emergency system” had the strongest relationship with performance in managing SRAs. This factor encompassed the arrangement of emergency medical service systems in the school setting, management of epinephrine auto-injectors and equipment, record document management, personal information security, confirmation of legal and guideline compliance for anaphylaxis, and self-development of up-to-date skills, among other aspects. Previous studies have reported that building an efficient and effective medical emergency response system enables all stakeholders to more effectively work with the responders in their communities and allows students to slow the progression of allergic reactions, helping them to receive further medical treatment [12,25,26]. Other research has indicated that school nurses emphasize the importance of proper documentation to improve understanding of the causative allergen, location, and management of allergic reactions in schools [12]. However, it has been suggested that a system for documenting all allergic reactions can help school nurses establish appropriate management and treatment strategies. In this study, having less than five years of career experience had a significant negative impact on performance in SRAs’ management. Furthermore, IPA results showed that when the working career was under five years, the factor “preparing to provide emergency care” was placed in quadrant II. Therefore, “preparing to provide emergency care” needs to be developed through CE to help elementary school nurses, especially those with less than five years of career experience, focus on improving their performance.

The factor “providing emergency care” showed the highest performance level, as well as the highest coefficient of determination. Moreover, the four items with the highest determinant coefficients all belonged to the factor “providing emergency care”. These items were: “when the airway is not clear, administer an epinephrine injection immediately”, “when the airway is not clear, lead in performing cardiopulmonary resuscitation, if needed”, “teach SRAs how to self-administer their epinephrine medication”, and “when the airway is clear, remove the triggering cause”. This suggests that elementary school nurses need training to improve their skills in administering epinephrine injections so they can teach SRAs to self-inject. In particular, although completing a CPR training course showed the highest performance level of all items, elementary school nurses still required additional training in taking a leading role in CPR among school staff. Education for school nurses on CPR could include not only basic life support certification but also frequent mock-emergency-scenario training sessions [21]. In the IPA results, among nurses for whom clinical information had not been provided, the factor “providing emergency care” was included in quadrant II, indicating that improvement efforts should be focused on performance. The primary channel for providing clinical information is through CE. For elementary school nurses, it is crucial to maintain up-to-date nursing skills to provide emergency care and instruct SRAs in self-management. Therefore, regular CE programs consisting of practical emergency nursing skills should be provided to elementary school nurses, along with the necessary clinical information.

Previous studies have highlighted the critical importance of implementing training programs for school nurses, as nearly half of students with food allergies may not have an anaphylaxis management plan in place [27]. Similarly, our study, which used IPA, found that the factors “utilizing individualized healthcare plans”, “conducting health education and staff training”, and “evaluating the student” were included in quadrant III, indicating that elementary school nurses underappreciated these areas. However, among elementary school nurses who agreed that schools need to be allowed to purchase epinephrine auto-injectors under law, “utilizing individualized healthcare plans”, “conducting health education and staff training”, and “evaluating the student” were plotted in quadrant II, indicating a need for intensive improvements. Furthermore, those who agreed with this statement demonstrated a significantly higher performance level compared to those who did not agree. In the United States, most states have laws protecting schools’ ability to purchase and stock epinephrine auto-injectors for emergency situations [20,28]. In contrast, South Korean law protects school staff who administer epinephrine but does not allow schools to purchase epinephrine auto-injectors [29]. It is possible that those who believe schools should be allowed by law to purchase epinephrine auto-injectors may have greater self-efficacy for anaphylaxis management [11] and be more aware of the importance of the three aforementioned areas, thus requiring intensive improvements. As a result, targeted programs should be developed for elementary school nurses, focusing on the areas of “utilizing individualized healthcare plans”, “conducting health education and staff training”, and “evaluating the student”, to strengthen their self-efficacy and improve their performance.

Although school nurses underestimated the factor “utilizing individualized healthcare plans”, it showed the second highest coefficient of determination, indicating a need for training. This factor encompasses various aspects, such as establishing an anaphylaxis management plan (AMP) based on a doctor’s prescribed action plan, providing the AMP before off-site activities, sharing information with school staff, supporting the development of an anaphylaxis alliance, and fostering collaboration. The item “collaborate with the principal, guardians, SRAs, class teachers, school nutritionists, and other related school staff to build alliances, and take the lead in these alliances” showed the third lowest performance level from all job items. Many state departments of education in various countries recommend establishing individualized healthcare plans to serve as guidelines for managing SRAs. Despite this recommendation, nearly half of students with a known history of anaphylaxis and who had experienced a reaction did not have an anaphylaxis action plan on file [30]. Another study reported that 87% of elementary school nurses used individualized anaphylaxis healthcare plans as a preventive management strategy [31]. A system for preparing individualized healthcare plans helped school nurses to perform the “evaluating the student” factor [24]. Evaluations should be conducted based on appropriate management and treatment strategies [12]. Therefore, elementary school nurses need to utilize individualized healthcare plans as a tool for building alliances with stakeholders, guiding the education of classmates, training school staff, evaluating strategies, and ultimately, optimizing the care of SRAs [12].

Since it is impossible to prevent all causative allergens that trigger anaphylaxis, it is vital to educate classmates and train school staff to raise awareness about recognizing the reaction and responding in emergency situations [23]. Education programs implemented in previous studies have been shown to improve knowledge, self-efficacy, and skills in administering epinephrine among school staff and students [18,19]. Another previous study reported that an education program related to asthma for students and staff increased the quality of life for students with asthma [32]. However, in this study, the job items that showed the lowest performance scores were “conduct staff training on administering an epinephrine auto-injector” and “prepare a health education teaching plan for students with anaphylaxis”. Timely administration of epinephrine can be life-saving and can buy time for further medical treatment [25]. School staff should be trained to recognize the progression of allergic reactions and to promptly administer epinephrine within the required timeframe. Simulation training for school staff using developed scenarios requiring anaphylaxis treatment is recommended as a training method [24]. According to a prior study, anaphylaxis events occurred most often in the classroom. Therefore, in addition to school staff, health education should be provided to classmates to improve their understanding of the disease and make them aware of their role in emergency situations. Health education plans should be intentionally prepared based on the developmental stage of elementary students and, consequently, be addressed in school policy in order to help decrease allergic reactions occurring in schools [24,33].

In this study, the IPA revealed that the factor “offering psychological support” was placed in quadrant II, indicating that intensive improvement is needed in this area. The factor “offering psychological support” was included in quadrant III (low importance and low performance) among elementary school nurses with less than five years of career experience, those who had not received clinical information, and those who disagreed that schools need to be allowed by law to purchase epinephrine auto-injectors. However, the perception of this factor’s importance changed to quadrant I (high importance and high performance) among elementary school nurses with more than five years of career experience and those who agreed that schools need to be allowed by law to purchase epinephrine auto-injectors. It shifted to quadrant II (high importance and low performance) among those who had received clinical information. The contents of this factor include providing post-incident emotional support to SRAs, offering counseling, referring to a school nutritionist for counseling, and referring to a school counselor for psycho-counseling, among other things. SRAs may feel angry and concerned with a fear about what might happen to them. High-risk children are exposed to psychological impacts that require close attention [9]. Previous study results showed that children and adolescents felt discriminated against at school because of severe allergies and stated that they felt embarrassed after experiencing allergic reaction episodes [34]. This indicates that school nurses must strive to minimize students’ feelings of being different from others and create a supportive environment for SRAs to promote their psychological health [28]. Therefore, elementary school nurses who have less than five years of experience as a school nurse, are not provided with clinical information, and disagree that schools should be allowed to purchase epinephrine auto-injectors under the law need to be aware of the importance of providing psychological support to SRAs. Particularly for those who are already aware of its importance through the clinical information that they had received, intensive improvement is necessary.

This study had several limitations. First, it did not assess the knowledge, attitudes, and skills of elementary school nurses in successfully managing the health of SRAs. Second, although this was a nationwide study, the sample size was small. Third, the authors did not compare participants working in full-time and part-time positions using the IPA method. Fourth, all subjects were from Korea; therefore, additional research on other ethnicities and countries is needed to generalize these findings. Nonetheless, this study is valuable due to its unique approach to evaluating standardized job competencies through the IPA method, identifying priority areas for consideration in CE.

## 5. Conclusions

This study assessed the competencies of elementary school nurses in managing the health of SRAs using standardized job competency measures. It also identified items and areas that should be incorporated into ongoing training curricula for elementary school nurses.

In this study, the elementary school nurse played a crucial role in the assessment and management of life-threatening emergencies. However, improvement efforts were needed for those who: (1) had less than five years of experience as a school nurse in emergency administration, (2) had not been provided with clinical information for emergency care, (3) had been provided with clinical information for psychological support, and (4) agreed that schools should be allowed to purchase epinephrine auto-injectors under the law for individualized healthcare plans, health education, and staff training. Therefore, when planning continuing education strategies to enhance the ability of elementary school nurses to manage severe-risk allergies, factors such as the nurse’s work experience, exposure to clinical information, and perception of the legality of purchasing epinephrine auto-injectors should be considered.

This study holds significant value, as it offers fundamental data for the development of a training curriculum for school nurses to manage the health of SRAs.

## Figures and Tables

**Figure 1 healthcare-11-02102-f001:**
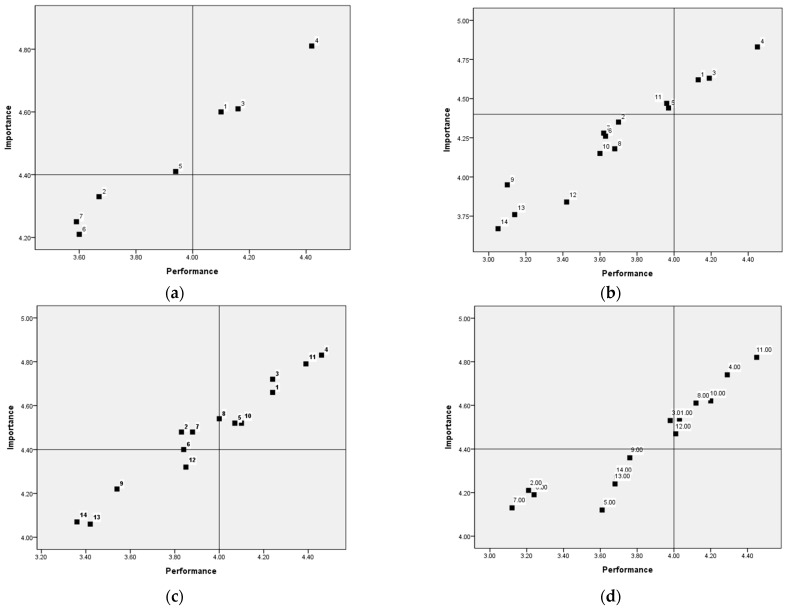
Areas that should be prioritized for training elementary school nurses to manage the health of students at risk for anaphylaxis. (**a**) Managing the health of students at risk for anaphylaxis, 1~7. (**b**) Have been provided clinical information, 1~7, and have not been provided clinical information on anaphylaxis, 8~14. (**c**) Schools need to be allowed by law to purchase epinephrine auto-injectors, 1~7, and schools do not need to be allowed by law to purchase epinephrine auto-injectors, 8~14. (**d**) Less than 5 years of career experience, 1~7, and more than 5 years of work experience, 8~14. Note: 1, 8 = assessing priority issues; 2, 9 = utilizing individualized healthcare plans; 3, 10 = preparing a school emergency system; 4, 11 = providing emergency care; 5, 12 = offering psychological support; 6, 13 = conducting health education and staff training; 7, 14 = evaluating the student.

**Table 1 healthcare-11-02102-t001:** Differences in performance levels based on the characteristics of the participants.

Variables		n (%)	Mean ± SD	t/F	*p*
Age	<30	15 (9.0)	3.67 ± 0.74	2.24	0.085
(years)	30–39	19 (11.4)	4.03 ± 0.59		
	40–49	67 (40.4)	4.14 ± 0.55		
	≥50	65 (39.2)	3.94 ± 0.79		
			45.77 ± 8.85		
Education	College	8 (4.8)	4.01 ± 0.75	1.25	0.287
	University	119 (71.7)	3.96 ± 0.70		
	Graduate school	39 (23.5)	4.16 ± 0.63		
Career experience as a hospital nurse	<1	25 (15.1)	4.05 ± 0.68	0.38	0.764
(years)	1–3	41 (24.7)	3.95 ± 0.62		
	3–5	58 (34.9)	3.95 ± 0.78		
	≥5	42 (25.3)	4.07 ± 0.66		
			4.05 ± 3.87		
Career experience as a school nurse	<5	29 (17.5)	3.75 ± 0.63 ^a^	2.85	0.039
(years)	5–14	37 (22.3)	4.19 ± 0.58 ^b^	a < b,c,d	
	15–24	56 (33.7)	4.10 ± 0.68 ^c^		
	≥25	44 (26.5)	3.91 ± 0.77 ^d^		
			16.41 ± 9.64		
Number of classes	<19	54 (32.5)	3.95 ± 0.75	0.41	0.663
	19–36	82 (49.4)	4.05 ± 0.70		
	≥37	30 (18.1)	3.99 ± 0.50		
			25.87 ± 13.61		
Previous anaphylaxis event in the	Yes	71 (42.8)	4.05 ± 0.73	0.72	0.471
community	No	95 (57.2)	3.97 ± 0.65		
Provided clinical information	Yes	157 (94.6)	4.04 ± 0.68	2.28	0.023
	No	9 (5.4)	3.50 ± 0.69		
Channel through which	Leaflet	15 (9.0)			
information was provided	CE	119 (71.7)			
	Internet	23 (13.9)			
Agreed that schools should be allowed to purchase epinephrine	Yes	72 (43.4)	4.14 ± 0.61	2.14	0.033
auto-injectors under law	No	94 (56.6)	3.91 ± 0.72		
All subjects were women					

**Table 2 healthcare-11-02102-t002:** Performance levels and determinant coefficients for managing the health of students at risk for anaphylaxis.

Items		Performance	Determination Coefficient
		Mean ± SD	
F1. Assessing priority issues		4.10 ± 0.80	15.32 ± 4.99
F1-1. Identify SRAs through a health status survey.		4.48 ± 0.96	14.04 ± 6.69
F1-2. Maintain an updated list of SRAs.		4.00 ± 1.03	14.88 ± 6.34
F1-3. Secure an action plan from the guardian, completed by a medical doctor, for health needs assessment.		3.82 ± 1.23	17.21 ± 6.15
F1-4. Obtain written consent from guardians for sharing personal information and providing first aid.		4.29 ± 1.04	15.31 ± 6.80
F1-5. Evaluate the self-care proficiency of SRAs.		4.13 ± 1.07	16.78 ± 6.26
F1-6. Ascertain the present state of EMSS and alliances.		3.88 ± 1.09	14.84 ± 6.25
F1-7. Assess the availability of anaphylaxis management resources such as medicine, equipment, and budget.		3.94 ± 1.10	13.98 ± 5.92
F1-8. Develop an emergency care plan.		4.22 ± 1.02	15.33 ± 6.13
F2. Utilizing individualized healthcare plans		3.66 ± 0.89	15.62 ± 4.64
F2-1. Incorporate the doctor-prescribed anaphylaxis action plan when establishing the AMP.		3.68 ± 1.24	16.88 ± 5.88
F2-2. Ensure easy accessibility of each individual’s first aid kit from various school areas during AMP formulation.		3.49 ± 1.25	16.86 ± 6.08
F2-3. Develop strategies in the AMP to minimize the risk of allergen exposure in school and out-of-class environments.		3.56 ± 1.16	16.87 ± 6.19
F2-4. Regularly review the AMP with the guardian considering the doctor’s medical advice.		3.45 ± 1.20	16.23 ± 5.94
F2-5. Review and provide the AMP before any off-site activities.		3.84 ± 1.11	15.24 ± 6.02
F2-6. Inform everyone about the location of the individual anaphylaxis first aid kit.		3.91 ± 1.15	13.75 ± 6.18
F2-7. Communicate any changes to school staff, students, guardians, and medical doctors.		3.83 ± 1.12	15.93 ± 5.77
F2-8. Provide comprehensive information to substitute teachers in the absence of the school nurse.		3.77 ± 1.11	15.95 ± 6.11
F2-9. Collaborate with the principal, guardians, SRAs, class teachers, school nutritionists, and other related school staff to build alliances, and take the lead in these alliances.		3.37 ± 1.24	15.71 ± 5.57
F2-10. Jointly decide on the storage location for the emergency kit.		3.67 ± 1.19	13.26 ± 6.07
F2-11. Collaboratively implement the AMP and support SRAs for a worry-free school life.		3.72 ± 1.12	15.62 ± 6.30
F2-12. Keep a record of the epinephrine auto-injector being checked out from and returned to storage in the IAHP.		3.65 ± 1.29	14.96 ± 6.42
F3. Preparing a school emergency system		4.16 ± 0.78	14.89 ± 4.95
F3-1. Set up a hospital contact system for emergencies.		4.41 ± 0.88	14.45 ± 6.40
F3-2. Establish a guardian notification system for emergencies.		4.42 ± 0.90	14.21 ± 6.40
F3-3. Create a school report system for emergencies.		4.28 ± 0.94	13.79 ± 6.38
F3-4. Administer prescribed medicine received from guardians and promptly replace it after use.		3.98 ± 1.16	16.44 ± 6.35
F3-5. Order and administer non-prescribed allergy medicine in the school health room.		4.41 ± 0.96	13.52 ± 7.05
F3-6. Acquire and manage equipment such as pulse oximeters and portable oxygen ventilators in the school health room.		4.40 ± 0.90	14.46 ± 6.80
F3-7. Regularly ensure that the prescribed medicine is not expired, replacing it one month before the expiration date.		4.05 ± 1.15	15.28 ± 6.77
F3-8. Consistently check that the portable oxygen ventilator is functioning properly.		4.01 ± 1.10	14.27 ± 6.42
F3-9. Ensure the anaphylaxis first aid kit storage is always unlocked.		4.21 ± 1.01	12.57 ± 6.65
F3-10. Include oral medicine, epinephrine auto-injector, action plan, and identification badge in the individual anaphylaxis first aid kit.		3.62 ± 1.23	14.89 ± 6.18
F3-11. Record the administration of medication, frequency, time, and date of allergic reactions accurately.		4.06 ± 1.11	15.59 ± 6.67
F3-12. Record emergency nursing care information in IAHP and the emergency sheet under the 5W1H principle.		3.95 ± 1.13	15.51 ± 6.40
F3-13. File documents pertaining to AMP, IAHP, emergency plans, etc.		3.96 ± 1.13	15.01 ± 6.44
F3-14. Uphold the confidentiality and privacy of SRAs.		4.20 ± 1.09	14.29 ± 6.57
F3-15. Verify legal and ethical criteria and guidelines for anaphylaxis.		4.09 ± 1.01	15.03 ± 6.75
F3-16. Complete a course in cardiopulmonary resuscitation training.		4.65 ± 0.82	14.30 ± 7.17
F3-17. Practice clinical skills for anaphylaxis emergency care.		4.06 ± 1.14	16.84 ± 6.81
F3-18. Stay informed about the latest medical information on anaphylaxis.		4.09 ± 1.04	16.65 ± 6.49
F4. Providing emergency care		4.42 ± 0.77	16.15 ± 5.60
F4-1. When the airway is clear.	4-1-1. Remove the triggering cause.	4.30 ± 0.97	17.93 ± 6.12
	4-1-2. Document the time lapse from exposure to symptom onset.	4.38 ± 0.93	15.43 ± 6.80
	4-1-3. Position the student sitting up and assist them in breathing.	4.47 ± 0.92	14.24 ± 6.83
	4-1-4. Monitor the student’s respiratory rate, level of consciousness, and oxygen saturation.	4.51 ± 0.89	14.57 ± 6.89
	4-1-5. Supervise the student as they take their prescribed medication.	4.46 ± 0.97	15.80 ± 6.56
	4-1-6. Provide emotional support to the anxious student.	4.45 ± 0.91	14.54 ± 6.78
	4-1-7. Contact the guardian.	4.56 ± 0.89	14.91 ± 6.95
	4-1-8. Monitor the student until symptoms subside or until the guardian arrives.	4.54 ± 0.89	15.70 ± 7.06
F4-2. When the airway is not clear.	4-2-1. Instantly dial 119 and initiate the EMSS.	4.53 ± 0.96	16.93 ± 7.09
	4-2-2. Administer an epinephrine injection immediately.	3.92 ± 1.35	20.10 ± 5.82
	4-2-3. Assess heart rate, respiratory rate, level of consciousness, and monitor oxygen saturation.	4.50 ± 0.92	16.77 ± 6.96
	4-2-4. Provide oxygen if necessary.	4.40 ± 0.94	17.30 ± 6.61
	4-2-5. Position the SRAs in a supine position, with lower extremities elevated.	4.37 ± 1.06	15.22 ± 7.19
	4-2-6. Lead in performing cardiopulmonary resuscitation, if needed.	4.44 ± 0.99	17.68 ± 7.21
	4-2-7. Transport the SRAs to the guardian-appointed hospital and notify the guardian.	4.51 ± 0.96	15.81 ± 7.31
	4-2-8. Communicate the emergency details to the hospital medical staff.	4.55 ± 0.95	15.23 ± 7.41
F4-3. Instruct SRAs to avoid exposure to the trigger allergen.		4.36 ± 0.89	16.04 ± 6.59
F4-4. Guide SRAs on how to take their self-carried prescribed medication during an allergic reaction.		4.45 ± 0.82	15.83 ± 6.86
F4-5. Teach SRAs how to self-administer their epinephrine medication.		4.18 ± 1.03	17.62 ± 6.67
F4-6. Advise SRAs to carry their prescribed medication at all times.		4.34 ± 0.91	15.86 ± 6.53
F4-7. Carry out cardiopulmonary resuscitation training.		4.51 ± 0.93	16.18 ± 6.61
F5. Offering psychological support		3.94 ± 0.90	14.37 ± 5.48
F5-1. Offer emotional support to SRAs for a healthier school experience.		4.01 ± 1.06	14.24 ± 6.37
F5-2. Arrange or provide post-incident support for SRAs displaying symptoms.		4.12 ± 1.03	15.06 ± 6.61
F5-3. Deliver health counseling to SRAs and their guardians.		4.27 ± 0.89	14.70 ± 6.30
F5-4. Refer SRAs and their guardians to a school nutritionist for counseling when needed.		3.83 ± 1.11	14.18 ± 5.85
F5-5. Refer SRAs to the school counselor for psychotherapy when necessary.		3.47 ± 1.21	13.67 ± 5.96
F6. Conducting health education and staff training		3.60 ± 0.89	14.96 ± 4.46
F6-1. Develop a health education plan.		3.92 ± 1.12	14.56 ± 6.15
F6-2. Formulate a staff training plan.		3.68 ± 1.12	15.40 ± 5.94
F6-3. Establish a plan for resource input, such as medicine, equipment, and budget.		3.77 ± 1.16	14.03 ± 5.75
F6-4. Advise SRAs to wear identification badges when in hazardous environments.		3.65 ± 1.24	15.22 ± 5.86
F6-5. Prepare a health education teaching plan for students with anaphylaxis.		3.28 ± 1.22	13.73 ± 5.76
F6-6. Provide education to raise awareness about allergies and anaphylaxis.		3.47 ± 1.24	14.02 ± 5.82
F6-7. Highlight the importance of each student’s role in emergency situations to foster peer support.		3.48 ± 1.23	14.78 ± 5.80
F6-8. Disseminate information about allergies and anaphylaxis through the school newsletter or website.		3.89 ± 1.11	12.49 ± 5.83
F6-9. Organize staff training on AMP.		3.53 ± 1.26	15.75 ± 5.93
F6-10. Conduct staff training on administering an epinephrine auto-injector.		3.17 ± 1.43	16.95 ± 6.29
F6-11. Train on how to respond within the critical four-minute window during an emergency situation.		3.73 ± 1.22	17.15 ± 6.05
F7. Evaluating the student		3.58 ± 0.95	14.35 ± 5.45
F7-1. Evaluate the function of the EMSS under emergency conditions.		3.66 ± 1.06	14.40 ± 5.88
F7-2. Evaluate the effectiveness of the anaphylaxis alliance based on the annual plan.		3.44 ± 1.07	14.13 ± 5.95
F7-3. Evaluate the provision of school resources such as medicine, equipment, and budget.		3.49 ± 1.09	13.95 ± 5.97
F7-4. Evaluate the efficacy of emergency care treatment under emergency conditions.		3.83 ± 1.06	15.18 ± 6.31
F7-5. Evaluate improvements in the self-care abilities of SRAs, as implemented by the IAHP.		3.72 ± 1.07	14.55 ± 6.12
F7-6. Evaluate whether in-school student health education on anaphylaxis was conducted as planned.		3.55 ± 1.13	13.89 ± 5.90
F7-7. Evaluate whether staff training was carried out as planned.		3.40 ± 1.11	14.28 ± 6.09

F = Factor; SRAs = students at risk for anaphylaxis; AMP = anaphylaxis management plan; EMSS = Emergency Medical Service System; IAHP = individualized anaphylaxis healthcare plan; 5W1H = who, what, when, where, why, and how.

**Table 3 healthcare-11-02102-t003:** Correlations between factors of performance, importance, and difficulty.

Variables	Performance	P-F1	P-F2	P-F3	P-F4	P-F5	P-F6	P-F7
I-F1	0.39 **	0.41 **	0.34 **	0.36 **	0.23 **	0.32 **	0.36 **	0.30 **
I-F2	0.48 **	0.41 **	0.49 **	0.46 **	0.24 **	0.46 **	0.46 **	0.35 **
I-F3	0.53 **	0.44 **	0.45 **	0.58 **	0.30 **	0.52 **	0.46 **	0.36 **
I-F4	0.43 **	0.35 **	0.27 **	0.47 **	0.38 **	0.35 **	0.31 **	0.26 **
I-F5	0.44 **	0.30 **	0.39 **	0.38 **	0.25 **	0.60 **	0.44 **	0.32 **
I-F6	0.44 **	0.37 **	0.35 **	0.40 **	0.18 *	0.44 **	0.59 **	0.35 **
I-F7	0.45 **	0.37 **	0.36 **	0.38 **	0.24 **	0.46 **	0.48 **	0.48 **
D-F1	0.12	0.25 **	0.09	0.09	0.07	0.04	0.11	0.09
D-F2	0.09	0.23 **	0.03	0.02	0.09	0.01	0.08	0.07
D-F3	0.07	0.19 *	0.01	−0.01	0.10	0.01	0.08	0.04
D-F4	0.06	0.20 **	0.03	0.01	0.06	−0.01	0.09	0.05
D-F5	0.07	0.16 *	0.01	−0.01	0.09	0.01	0.07	0.14
D-F6	0.11	0.23 **	0.04	0.01	0.14	0.05	0.10	0.08
D-F7	0.08	0.22 **	−0.01	0.06	0.13	−0.01	0.06	0.01

* *p* < 0.05, ** *p* < 0.001. Note: P = performance; I = importance; D = difficulty; F1 = assessing priority issues; F2 = utilizing individualized healthcare plans; F3 = preparing a school emergency system; F4 = providing emergency care; F5 = offering psychological support; F6 = conducting health education and staff training; F7 = evaluating the student.

**Table 4 healthcare-11-02102-t004:** Predictive variables for the performance level.

Variables	B	SE	Β	T	*p*
Constant	0.65	0.42		1.53	0.127
Importance of F3	0.73	0.09	0.52	8.03	<0.001
Career experience as a school nurse (d) = 1(<5 years)	−0.24	0.11	−0.13	−2.03	0.043

F = 35.63, *p* < 0.001, adj. R2 = 0.296. Note: F3 = preparing a school emergency system.

## Data Availability

Not applicable.

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
