# Peer review of "Evaluating Standardized Job Competencies for Managing Students at Risk for Anaphylaxis in Elementary School Nurses"

_healthcare, 2023, doi:10.3390/healthcare11142102_

Round 1
Reviewer 1 Report
This paper contains important results regarding children's health at school. There are some suggestions for improvement:
1) In line 61, IPA should be explained as to how it works as a technique.
2) What were the qualifications for the seven focus group members who developed the topics for the survey?
3) The brainstorming method suggested by the National Competency Standard should be explained.
4) By what criteria were the 14 items removed to get 82 survey items?
5) Example items for each of the factors should be given after line 127. Cronbach's alphas for each factor should be given as well.
6) Lines 133-4 mention performance, importance, and difficulty. How were these three concepts measured with the survey factors?
7) In lines 138-9, it states that variables with significant differences were converted to dummy variables for the regression. Dummy variables discard variance because they are categorical. Why were the original variables not used?
8) The IPA method should be explained in detail after line 141. How is it determined which items go in which quadrant?
9) In Section 2.4, how did branches recruit participants (i.e, with emails, letters, etc.)? What are SNS networks?
10) In Section 3.1, it states that less than half of the nurses had prior experience working with anaphylaxis. Thus, they may not know what is important in such situations. This should be listed as a limitation.
11) Is there demographic data to report on gender, education, and/or marital status of school nurses?
12) In Section 3.2, it mentions coefficients of determination. This statistic is typically the square of the correlation coefficient and has values of less than 1. What statistic is being referenced here?
13) In the first paragraph of Section 3.2, it mentions factors, but more than 7 factors are listed. I assume the "factors" listed are items?
14) In Section 3.5, how were items chosen for quadrants? - were there statistical values that determined their placement? These should be reported. Also, the items listed only represent Quadrants II and III, where items in Quadrants I and IV are mentioned in the Discussion. There should be a table that shows where all items for all types of nurses are placed.
15) In line 348, it states that "all areas" are important, but just because the factor was correlated with importance and difficulty, all items may not have been correlated with them.
16) In two places in the Discussion, the values for the coefficient of determination are explained as showing that the nurses "need training" in specific areas. How does the coefficient of determination reflect a need for training?
17) The limitation of generalizability to other ethnicities and geographic areas should be mentioned.
Author Response
Dear Dr.
We thank referees for careful reading our manuscript and for giving useful comments. We have revised the manuscript ID. 2500615 on the basis of the reviewer's comments.
I hope that the revised paper meets your approval and will be more suitable for publication in Healthcare.
Our responses to the reviewer’s comments are as follows:
1) In line 61, IPA should be explained as to how it works as a technique.
-> Thank you for your suggestion. There are explanation of utility of IPA in the introduction. “analysis (IPA) can be a valuable method [13]. This approach aids in identifying problematic areas for strategically developing training curricula and improving the quality of health services”
And the technique is described in the data analysis “A two-dimensional IPA model was created, with a mean importance level of 4.46 and a mean performance level of 4.01. The model was divided into four quadrants, with performance on the x-axis and importance on the y-axis. Quadrant I contain attributes that are perceived as highly important with a high level of performance. School nurses should be encouraged to maintain their excellent work in these areas. Quadrant II includes factors that are considered very important, but with a relatively low performance level. Improvement efforts should be focused on enhancing performance in these areas. Quadrant III consists of attributes that are perceived as having low importance and low performance. These factors should be considered a low priority for training. Lastly, Quadrant IV contains attributes of low importance but relatively high performance. In these situations, some attributes may be overemphasized, indicating that current practices may be unnecessary.”
Thank you very much for your comment. It helped us in improving the study.
2) What were the qualifications for the seven focus group members who developed the topics for the survey?
-> Thank you for your suggestion. We have added the sentence and the reference “(4 school nurses, 1 emergency room nurse, 1 university professor in nursing faculty, and 1 nurse expertise in job analysis)”
3) The brainstorming method suggested by the National Competency Standard should be explained.
->Thank you for your suggestion. We have revised the sentence “. These members convened 10 times for 180 minutes each to develop job tasks associated with the elementary school nurses in managing the health of SRAs based on consensus [17].”
4) By what criteria were the 14 items removed to get 82 survey items?
-> Thank you for your suggestion. We revised the sentence to make it clear “Fourteen tasks that did not meet the standard content validity index (.80) were removed. To confirm the underlying structure of the rest of the tasks, factor analysis was conducted, resulting in the extraction of seven factors with a cumulative percentage of 68.6. No factors were eliminated. Consequently, 82 items of standardized job competency for elementary school nurses in managing the health of students at risk for anaphylaxis were developed.”
5) Example items for each of the factors should be given after line 127. Cronbach's alphas for each factor should be given as well.
-> Thank you for your suggestion. All the items for each factor are already suggested in table 2.
We are submitting appendix for Cronbach's alphas for each factor.
6) Lines 133-4 mention performance, importance, and difficulty. How were these three concepts measured with the survey factors?
-> Thank you for your suggestion. In L117, the Likert scale has been described. “Participants rated each task on a 5-point Likert scale, ranging from 5 (very highly performed, very highly important, very highly difficult) to 1 (minimally performed, minimally important, minimally difficult).”
7) In lines 138-9, it states that variables with significant differences were converted to dummy variables for the regression. Dummy variables discard variance because they are categorical. Why were the original variables not used?
-> Thank you for your suggestion. The working career as a school nurse, providing clinical information, allowing schools to purchase epinephrine auto-injectors under the law was suggested as categorical in table 1. Especially, providing clinical information, allowing schools to purchase epinephrine auto-injectors under the law are measured by dichotomous scale.
8) The IPA method should be explained in detail after line 141. How is it determined which items go in which quadrant?
-> Thank you for your suggestion. Items are determined by using SPSS program. We have added “All statistical analyses were performed using SPSS version 23.0 (IBM Corp., Armonk, NY, USA).” In the data analysis.
To give more information about IPA methods, we have input a reference number.
9) In Section 2.4, how did branches recruit participants (i.e, with emails, letters, etc.)? What are SNS networks?
-> Thank you for your suggestion. In L169, we explained how each branch recruit participants. But the authors confirmed by telephone whether they were in inclusion criteria. We have revised the sentence to make this clear.
10) In Section 3.1, it states that less than half of the nurses had prior experience working with anaphylaxis. Thus, they may not know what is important in such situations. This should be listed as a limitation.
-> Thank you for your suggestion. It was to find whether they had prior experienced events of anaphylaxis in the community rather than to find prior experience working with anaphylaxis. We have revised the sentence. “A total of 42.8% of the subjects had prior experienced events of anaphylaxis in the community.”
Thank you very much for your comment. It helped us in improving the study.
11) Is there demographic data to report on gender, education, and/or marital status of school nurses?
-> Thank you for your suggestion. We have added gender, education in table 1 and in the text. But we did not investigate marital status.
12) In Section 3.2, it mentions coefficients of determination. This statistic is typically the square of the correlation coefficient and has values of less than 1. What statistic is being referenced here?
-> determined coefficients were obtained by multiplying the mean values of importance and difficulty. It is mainly used in job analysis. The example of reference is below.
Kang HS, Son HM, Lim NY, Cho KS, Kwon SB, Yi YJ, et al. Job analysis of clinical research coordinators using the DACUM process. J Korean Acad Nurs. 2012;42(7):1027e38. http://dx.doi.org/10.4040/jkan.2012.42.7.1027
13) In the first paragraph of Section 3.2, it mentions factors, but more than 7 factors are listed. I assume the "factors" listed are items?
-> Thank you for your suggestion. First paragraph were about The factor with the highest coefficient of determination and the lowest performance score
Second paragraph were about items
14) In Section 3.5, how were items chosen for quadrants? - were there statistical values that determined their placement? These should be reported. Also, the items listed only represent Quadrants II and III, where items in Quadrants I and IV are mentioned in the Discussion. There should be a table that shows where all items for all types of nurses are placed.
-> Thank you for your suggestion. We have used factors for the IPA analysis. We have used SPSS program to find this result. The reason we discussed Quadrants I, Quadrants II and III are described in the discussion.
15) In line 348, it states that "all areas" are important, but just because the factor was correlated with importance and difficulty, all items may not have been correlated with them.
-> Thank you for your suggestion. We have revised the sentence. “This result suggests that elementary school nurses who perform comprehensive assessments of SRAs are more likely to perceive all areas as important.”
Thank you very much for your comment. It helped us in improving the study.
16) In two places in the Discussion, the values for the coefficient of determination are explained as showing that the nurses "need training" in specific areas. How does the coefficient of determination reflect a need for training?
-> determined coefficients were obtained by multiplying the mean values of importance and difficulty. It is mainly used in job analysis. The example of reference is below. The determinant coefficient is an index used to suggest training needs. High index items are developed as content in a continuous training curriculum. The reference is below.
Lee YJ, Choo J, Cho JH, Kim SN, Lee HE, Yoon SJ, et al. Development of a stan[1]dardized job description for healthcare managers of metabolic syndrome management program in Korean Community health centers. Asian Nurs Res. 2014;8:57e66. http://dx.doi.org/10.1016/j.anr.2014.02.003 13. Kang HS, Son HM, Lim NY, Cho KS, Kwon SB, Yi YJ, et al. Job analysis of clinical research coordinators using the DACUM process. J Korean Acad Nurs. 2012;42(7):1027e38. http://dx.doi.org/10.4040/jkan.2012.42.7.1027
17) The limitation of generalizability to other ethnicities and geographic areas should be mentioned.
-> Our study subjects are all Korean. And we have suggested the limitation of generalizability in L501 “Therefore, additional research is needed to generalize these findings.”
Thank you very much for your comment. It helped us in improving the study.

Reviewer 2 Report
Thank you very much for the opportunity to review this manuscript and I am glad that school nursing is a fact of life in Korea, what an envy!
I feel that the manuscript is very well written, the introduction does a good literature review. The methods are well written, the results are well described, perhaps table 2 could have a supporting figure for factors...it is a difficult table to read because of the size.
The discussion is comprehensive and addresses the large number of results. The conclusions are supported by the results and the references are in the correct format.
Congratulations to the authors
Author Response
Dear Dr.
We thank referees for careful reading our manuscript and for giving useful comments. We have revised the manuscript ID. 2500615 on the basis of the reviewer's comments.
I hope that the revised paper meets your approval and will be more suitable for publication in Healthcare.
Our responses to the reviewer’s comments are as follows:
- table 2 could have a supporting figure for factors...it is a difficult table to read because of the size.
- Thank you for your suggestion. We are sorry to give you difficulty in reading the table. We have cut the table into 3 pieces.
Reviewer 3 Report
This article gives interesting results about job competencies of elementary school nurses in managing the health of students at risk for anaphylaxis (SRAs) in South Korea. The topic is within the scope of Healthcare but needs to consider following comments for publication.
1. How about quantitatively presenting the important data obtained in this survey in the abstract? It seems that the method of factor analysis is used, but the method cannot be fully understood from the abstract.
2. I could not fully understand the content of Reference 20, National Competency Standards: Manual for job competency standard 2023. Would you mind adding a specific explanation?
3. With regard to providing emergency care in the consideration, it may be necessary to discuss the creation of a system such as having multiple people implement it rather than having it implemented by a single nurse. Shouldn't we be thinking about organizational systems that go beyond the capabilities of individuals?
Author Response
Dear Dr.
We thank referees for careful reading our manuscript and for giving useful comments. We have revised the manuscript ID. 2500615 on the basis of the reviewer's comments.
I hope that the revised paper meets your approval and will be more suitable for publication in Healthcare.
Our responses to the reviewer’s comments are as follows:
- How about quantitatively presenting the important data obtained in this survey in the abstract? It seems that the method of factor analysis is used, but the method cannot be fully understood from the abstract.
-> Thank you for your suggestion. As requested, we have added “Seven factors were obtained from the factor analysis with a cumulative percentage of 68.6.” in the abstract. We tried to present quantitatively data but the abstract paragraph is already 212 words.
- I could not fully understand the content of Reference 20, National Competency Standards: Manual for job competency standard 2023. Would you mind adding a specific explanation?
-> Thank you for your suggestion. Korea Job Competency Standards(Ministry of employment and Labor) have used Norton(2013) job analysis methods. Therefore, we revise the sentence to “These members convened 10 times for 180 minutes each to develop job tasks associated with the elementary school nurses in managing the health of SRA based on consensus [17].“ And we deleted the above reference and revised all the numbers of the reference.
Reference 17) Norton, RE.; Moser J. DACUM handbook. 4th ed. Columbus (OH): Center on Education and Training for Employment; The Ohio State University, 2013.
- it may be necessary to discuss the creation of a system such as having multiple people implement it rather than having it implemented by a single nurse. Shouldn't we be thinking about organizational systems that go beyond the capabilities of individuals?
-> Thank you for your suggestion. The tasks of factor organizational systems already contain your comment. To make it more clear, we have changed the renamed factor 3 into "Preparing a school emergency system". And we have revised “Previous studies have reported that building an efficient and effective medical emergency response system able all the stakeholders more effectively work with the responders in their communities and allow students to slow the progression of allergic reactions, enabling them to receive further medical treatment” to reflect your opinion.
Reviewer 4 Report
This study aimed to evaluate the standardized job competencies of elementary school nurses in managing the health of students at a risk of anaphylaxis. The authors report that when planning continuing education strategies to enhance the ability of elementary school nurses to manage severe risk allergies, factors such as nurses’ work experience, exposure to clinical information, and perception of the legality of purchasing epinephrine auto-injectors should be considered.
Overall, this work tackles an interesting topic and sheds new light on school nursing. The scientific rationale for the investigation is sound, and the authors present a few novel and important observations that provide a logical extension of previous studies on the standardized job competencies of elementary school nurses. The manuscript is well organized and has been carried out. However, some information is missing and should be provided to clarify some issues.
The following points should be addressed:
1. Could you provide an additional explanation about the current situation of continuing education for school nurses to perform health management for students at risk of anaphylaxis in the Introduction section?
2. Ethical considerations are included in the data collection section. I recommend that the author should describe ethical considerations and data collection sections separately.
3. You should explain why the sample size was small and the alternatives.
N/A
Author Response
Dear Dr.
We thank referees for careful reading our manuscript and for giving useful comments. We have revised the manuscript ID. 2500615 on the basis of the reviewer's comments.
I hope that the revised paper meets your approval and will be more suitable for publication in Healthcare.
Our responses to the reviewer’s comments are as follows:
1.Could you provide an additional explanation about the current situation of continuing education for school nurses to perform health management for students at risk of anaphylaxis in the Introduction section?
Thank you for your suggestion. We revised as you suggested.
“Other countries have conducted training programs for school nurses to improve on responding emergencies of students with life-threatening allergies [10-12] and distributed manuals as to serve as the guidelines [13-15]. In South Korea, although school nurses are now legally allowed to inject epinephrine auto-injectors in school setting [8], specific training curriculum of anaphylaxis for school nurse are rarely developed. They come across this disease when general health education is continued. Needless to say, standardized manuals are yet developed and distributed.”
2.Ethical considerations are included in the data collection section. I recommend that the author should describe ethical considerations and data collection sections separately.
Thank you for your suggestion. We revised as you suggested.
- You should explain why the sample size was small and the alternatives.
Thank you for your suggestion. We revised as you suggested. “Two hundred seventy school nurses volunteered for this study. The authors called to each school nurses to confirm whether they include in the study subjects and mailed questionnaires to 240 elementary school nurses who were in inclusion criteria. The authors also offered remind call to each school nurse to respond to the questionnaire.”
Round 2
Reviewer 1 Report
Several of the reviewer comments were answered adequately. However, there are some comments that still need to be addressed:
1) Stating that SPSS was used for the IPA analysis does not explain the criteria for how each item was placed in each quadrant. The method needs to be explained in more detail.
2) What SNS networks are was not explained.
3) No table was added with statistical values for each item from the IPA analysis.
4) In the first paragraph of Section 3.2, there are still more labels given than factors. It is unclear whether the labels are for items or not.
5) For generalizability to other ethnicities and countries, it needs to be stated that all subjects are from Korea to point out the generalizability limitation, not just that "additional research is needed."
There are some English grammar errors. An edit by a native English speaker is needed.
Author Response
Dear Dr.
We thank referees for careful reading our manuscript and for giving useful comments. We have revised the manuscript ID. 2500615 on the basis of the reviewer's comments.
I hope that the revised paper meets your approval and will be more suitable for publication in Healthcare.
Our responses to the reviewer’s comments are as follows:
1) In the first paragraph of Section 3.2, there are still more labels given than factors. It is unclear whether the labels are for items or not.
-> Thank you very much for your comment. First paragraph starts with [The factor with the highest determinant coefficients was……], and the second paragraph starts with [The item of job competency that showed…..]. even though, it still seems to be unclear. Therefore, in this section, we input code number of each factors and items accordance to table1.
2) For generalizability to other ethnicities and countries, it needs to be stated that all subjects are from Korea to point out the generalizability limitation, not just that "additional research is needed."
-> Thank you very much for your comment. We added “. Fourth, all subjects are from Korea therefore, additional research to other ethnicities and countries is needed to generalize these findings.” In L 501
3) There are some English grammar errors. An edit by a native English speaker is needed.
-> Thank you for your suggestion. Before we submitted the first version manuscript, we already received the editorial service from USA company. But there must have been other English grammar errors while revising the manuscript. We have undergone editorial service again to meet your request.
4) What SNS networks are was not explained.
-> Each branch had group network using Kakaotalk (kakao corperation, Korea). I would rather not input the specific SNS name.
5) Stating that SPSS was used for the IPA analysis does not explain the criteria for how each item was placed in each quadrant. The method needs to be explained in more detail.
No table was added with statistical values for each item from the IPA analysis.
-> Thank you very much for your comment. As we have already explained in former review answer, the criteria for how each item was placed in each quadrant is already described in the manuscript. “A two-dimensional IPA model was created, with a mean importance level of 4.46 and a mean performance level of 4.01. The model was divided into four quadrants, with performance on the x-axis and importance on the y-axis. Quadrant I contain attributes that are perceived as highly important with a high level of performance. School nurses should be encouraged to maintain their excellent work in these areas. Quadrant II includes factors that are considered very important, but with a relatively low performance level. Improvement efforts should be focused on enhancing performance in these areas. Quadrant III consists of attributes that are perceived as having low importance and low performance. These factors should be considered a low priority for training. Lastly, Quadrant IV contains attributes of low importance but relatively high performance. In these situations, some attributes may be overemphasized, indicating that current practices may be unnecessary.”
There are no statistical values. The results of IPA analysis is drawn in a graph with Quadrant axis. To give more information about IPA methods, a reference is below.
Martilla, J. A.; James, J. C. Importance-Performance Analysis. J Mark. 1977, 41 (1), 77–79.
Thank you very much for your comment. It helped us in improving the study.